# Using synchronized oscillators to compute the maximum independent set

Antik Mallick[1], Mohammad Khairul Bashar[1], Daniel S. Truesdell [1], Benton H. Calhoun[1], Siddharth Joshi [2] & Nikhil Shukla [1✉]

Not all computing problems are created equal. The inherent complexity of processing certain classes of problems using digital computers has inspired the exploration of alternate computing paradigms. Coupled oscillators exhibiting rich spatio-temporal dynamics have been proposed for solving hard optimization problems. However, the physical implementation of such systems has been constrained to small prototypes. Consequently, the computational properties of this paradigm remain inadequately explored. Here, we demonstrate an integrated circuit of thirty oscillators with highly reconfigurable coupling to compute optimal/near-optimal solutions to the archetypally hard Maximum Independent Set problem with over 90% accuracy. This platform uniquely enables us to characterize the dynamical and computational properties of this hardware approach. We show that the Maximum Independent Set is more challenging to compute in sparser graphs than in denser ones. Finally, using simulations we evaluate the scalability of the proposed approach. Our work marks an important step towards enabling application-specific analog computing platforms to solve computationally hard problems.

[1] Department of Electrical and Computer Engineering, University of Virginia, Charlottesville, VA 22904, USA. [2] Department of Computer Science and Engineering, University of Notre Dame, Notre Dame, IN 46556, USA. ✉email: ns6pf@virginia.edu

While digital computing—the work-horse of modern information technology—provides a powerful computing framework, different problems have different computational complexities. There are certain problems—commonly known as NP-hard problems—that are still considered intractable to solve using digital machines, and require exponentially increasing resources for increasing sizes of the input. Moreover, many practical problems in physics (e.g., ground-state problem of spin-glasses[1]), bioinformatics (e.g., protein folding[2], medical imaging[3]), combinatorial optimization (e.g., scheduling[4], traveling salesman problem[5]), circuit design (e.g., field programmable gate array (FPGA) routing[6]) among others belong to this class of computational complexity. This has naturally motivated the quest to explore beyond-digital computing fabrics[7–9] including dynamical systems to address this increasingly valuable class of problems.

In this work, we design and fabricate an integrated circuit (IC) of 30 relaxation oscillators with reconfigurable coupling to explore the properties of this dynamical system for computing the prototypically hard maximum independent set (MIS) problem. Combinatorial optimization problems like computing the MIS entail finding the optimal value of a function in a discrete or combinatorial domain set. Specifically, the MIS problem is NP-hard[10] implying that even the best deterministic algorithms and hardware implementations (including the dynamical system proposed here) may result in searching the entire high-dimensional solution space for at least some problem instances. It is believed that dynamical systems such as Hopfield networks[11], spiking neurons[12], cellular automata[13], coupled oscillators[14–16]—the focus of the present work, as well as other implementations[17–20] can search the solution in a highly parallel fashion[21], and hence, could potentially solve or approximate such problems efficiently.

However, the physical implementation of analog systems such as synchronized oscillators must contend with design challenges related to noise and stability; this has traditionally been a significant advantage for digital computing. Analog computing was explored in the 1950s[22,23] but was largely abandoned for digital information processing owing to its better noise immunity and the then relatively immature (analog) process technology[24]. However, with dramatic strides in process control and the inherently relevant computational properties of such systems[25–33], we revisit the analog approach for solving hard computational problems by using coupled oscillatory systems[34–37].

The concept of computing using oscillators has experienced renewed interest owing to the emerging device technologies that promise potentially compact oscillator implementations[38–45]. For instance, researchers recently demonstrated the ability to perform vowel classification using the frequency synchronization characteristics of four coupled spin transfer torque (STT) oscillators[46]. Further, J. Chou et al.[47] and T. Wang et al.[48] demonstrated the ability to compute the Max-Cut using LC oscillators. Similarly, other emerging technologies such as insulator–metal phase transition oxide (example, $VO_2$[49–52], $TaO_x$[53], $NbO_x$[54])-based oscillators have also been used to explore the computational properties of coupled oscillators. While these works are promising[55–57], the currently nascent nature of their underlying device technologies has constrained the system size and reconfigurability owing to inherent variability and limited process control[58–60]. Consequently, this has limited the experimental exploration and understanding of the coupled oscillator dynamics in larger systems. Therefore, in this work, we leverage the highly mature CMOS process technology to demonstrate a 30 relaxation oscillator platform with reconfigurable connectivity, and subsequently, characterize its ability to solve the prototypically hard maximum independent set (MIS) problem. In contrast to our earlier work with $VO_2$ oscillators[49], the larger oscillator system demonstrated

here using the CMOS-based Schmitt trigger oscillators uniquely enables us to evaluate the computing properties in a larger system as well as investigate the effect of various parameters such as average connectivity on the computational characteristics.

## Results

**Computing the MIS using coupled oscillators**. The MIS of a graph, G (V, E) (V: Vertices; E: Edges), is defined as the largest subset of nodes having no edges amongst them. The MIS problem is a prototypical combinatorial optimization problem with extensive applications in coding theory[61], resource allocation[62], molecular biology[63], and VLSI design[64] (Fig. 1). To compute the MIS using the coupled oscillators, we configure the oscillator network such that every node of the input graph is represented by an oscillator, and every edge by a coupling capacitor. As demonstrated in our prior work[49], this results in a unique circular phase ordering (Fig. 1) such that the vertices (nodes) belonging to an independent set of the graph appear adjacent to each other (see Supplementary Note 1). Subsequently, the phase ordering can be sorted into independent sets wherein the largest such set approximates the solution to the MIS. For the sample graph considered in Fig. 1, the bottom panel shows the experimentally observed time-domain waveform and the corresponding circular ordering (here, …1, 4, 6, 3, 2, 5, 1, 4…) of the oscillator phases. Subsequently, this phase order can be divided into independent sets: {1,4,6}, {3}, {2,5} by identifying adjacent nodes in the ordering that have an edge. The largest independent set {1,4,6} approximates the MIS.

**Coupled oscillator hardware**. We aim to experimentally explore the computational properties of the coupled oscillators over a broad range of network size, connectivity, and patterns. To facilitate this investigation, we develop an IC, using the bulk CMOS 65 nm technology, consisting of 30 programmable relaxation oscillators which can be capacitively coupled to each other in any arbitrary configuration, i.e. each oscillator can be coupled to any and all of the oscillators in the network (Fig. 1) (see Supplementary Note 2 for details of the IC). As discussed earlier, the mature CMOS technology enables us to characterize the close-to intrinsic computational characteristics of this analog-computing paradigm without being impeded by factors such as large variability and limited endurance. Each oscillator is implemented using a Schmitt trigger inverter where the oscillations are stabilized using negative RC feedback. Furthermore, each oscillator is equipped with a current starver circuit (implemented at the header and footer) to modulate the quiescent point and the operating frequency. The reconfigurable capacitive coupling architecture is implemented using a 30 line (=number of oscillators) bus along with transmission gate (T-gate)-based switches that facilitate programmable 'all-to-all' connectivity among the oscillators. This essentially enables us to map the adjacency matrix, A (which specifies the edges between the nodes) of any arbitrary graph (up to 30 nodes) directly on to the physical hardware. Other peripheral blocks of the oscillator platform include: (a) serial-in parallel-out (SIPO) registers to program to the oscillators, and coupling capacitors; (2) Schmitt trigger inverter-based hysteretic output buffer to digitize the output of each oscillator while preserving the phase and frequency information; (3) 32:1 multiplexer to read the (buffered) oscillator output; the MUX reduces the number of I/O pads required to measure the output. The output of one oscillator is read directly from the IC and is considered as a reference for comparing the phase.

To process a graph using this platform, the oscillators and the coupling elements are programmed to represent the nodes and

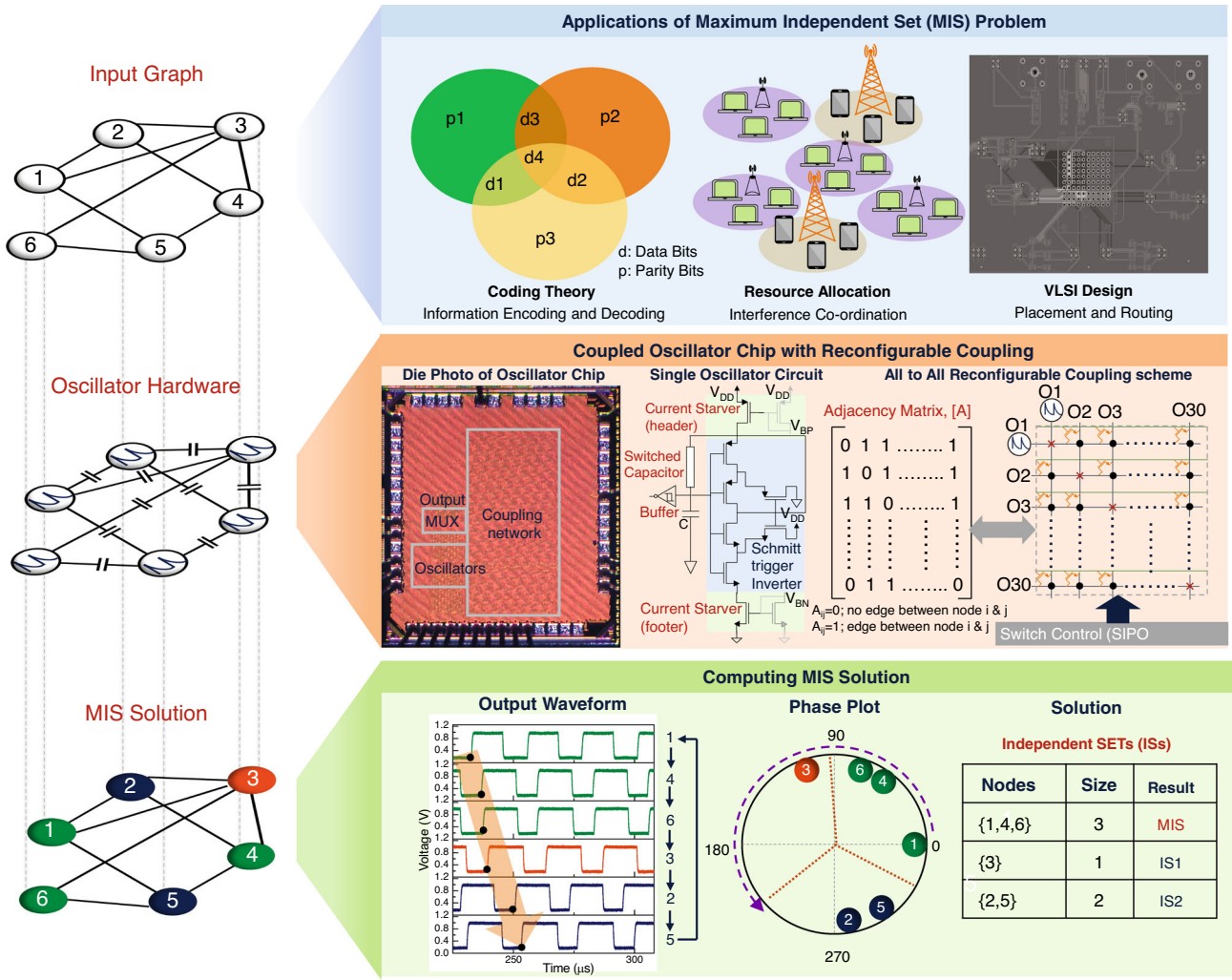

**Fig. 1 Reconfigurable coupled oscillator platform to solve maximum independent set (MIS).** (Top panel) Solving the computationally hard MIS problem in graphs and its practical applications; (Middle panel) Features of the IC: Die photo, oscillator circuit schematic, and illustration of the all-to-all capacitive coupling scheme implemented in the IC; (Bottom panel) Experimental time domain waveform, phase sequence, and the independent sets computed from the coupled oscillator dynamics. The largest independent set approximates the MIS.

the edges of the input graph, respectively; the number of oscillators corresponds to the number of rows (or columns) of $A$ (Fig. 1), whereas each element of the matrix $A_{ij}$ represents an edge between node $i$ and $j$; $A_{ij} = A_{ji} = 1$ denotes the presence of an edge, whereas $A_{ij} = A_{ji} = 0$, symbolizes the absence of an edge.

Therefore, each row of the matrix is formulated as a binary bitstream and passed onto the SIPO register for programming the coupling elements. Since the steady-state phase ordering of the oscillators encodes the solution, the phase of each oscillator is read (through the 32:1 MUX) by comparing it to the phase of the reference oscillator whose output is measured directly from the IC. Subsequent processing such as partitioning the circular ordering into independent sets is performed using software.

**Computational characteristics of the coupled oscillators.** We experimentally test randomly generated graph instances of varying size ($V = 5, 10, 15, 20, 25, 30$) and average connectivity ($\eta = 0.2, 0.4, 0.6, 0.8$); three graphs are tested for every combination of ($V, \eta$). Average connectivity ($\eta$) is defined as the ratio of the number of edges in the graph to the total number of edges in an all-to-all connected graph of the same size. Figure 2a shows bubble plots comparing the MIS solutions obtained using the

coupled oscillators, and the traditional Bron–Kerbosch (B–K) algorithm which guarantees an optimal MIS solution, when it converges. It can be observed that the solution to most of the analyzed graph instances lies near—or on the identity line (i.e. $y = x$) implying that the oscillator system computes an MIS that is close to the optimal solution computed by the B–K algorithm. Analysis shown in the inset of Fig. 2b reveals that the oscillators compute an optimal MIS solution in 71% of the tested graphs, while computing an approximate solution within 1 node of the optimal MIS in >90% of the cases. As discussed in the following sections, the reduction in accuracy primarily arises from the suboptimal nature of the solution in sparser graphs (low $\eta$) which are more challenging to solve than graphs with larger edge density—a fundamental property of the NP-hard MIS problem.

To attain further insights into how the size and the connectivity of the input graph affect the computational characteristics of the oscillators, we analyze the quality of the solution (quantified as the deviation $\delta$ (in %) from the optimal MIS solution) as a function of $V, \eta$. It can be observed from Fig. 3a that the oscillators tend to settle to a sub-optimal ordering, and thus, compute a sub-optimal MIS for sparser graphs with lower $\eta$, while optimal ordering is observed in denser graphs. This trend strongly aligns with the observed

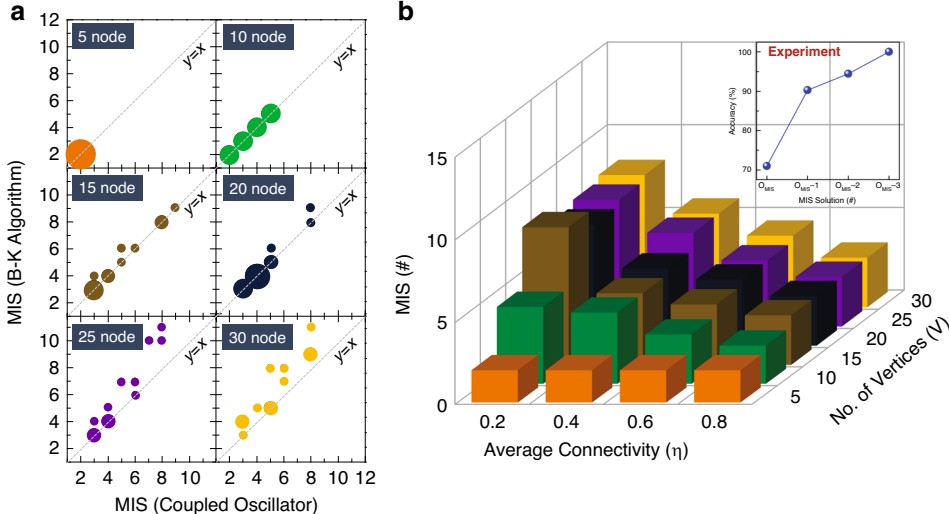

**Fig. 2 MIS solution computed by the coupled oscillators. a** MIS solution experimentally obtained from the coupled oscillators and its comparison with the optimal solution (computed using the B–K algorithm). **b** Evolution of size of the MIS (computed by the oscillators) as a function of graph size ($V$) and connectivity ($\eta$). The MIS solution increases with the size and the sparsity of the graph. The MIS for each ($V$, $\eta$) is averaged over the three measured random graph instances. Inset shows the overall accuracy as a function of the optimality of the MIS solution for the experimentally measured graph instances. $O_{MIS}$ represents optimal MIS solution, and $O_{MIS}$-$k$ is deviation from optimal MIS by $k$ ($k = 1,2,3$).

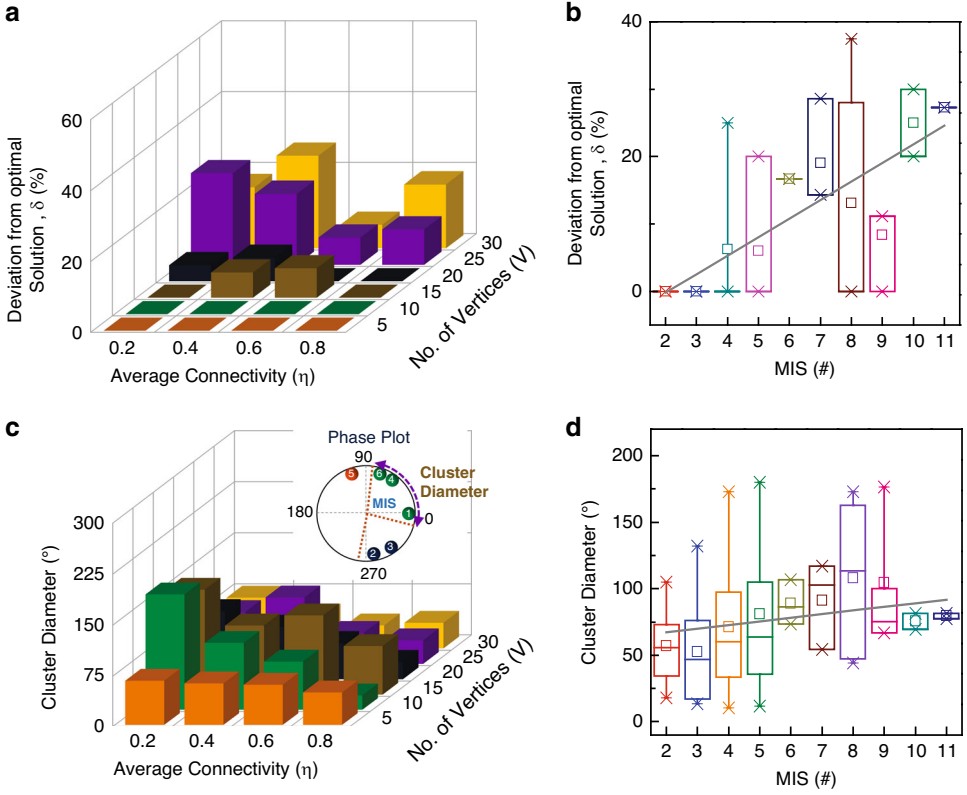

**Fig. 3 Computational characteristics of the coupled oscillators. a** Effect of graph size ($V$) and connectivity ($\eta$) on the quality of measured MIS solution (expressed as a deviation ($\delta$) from the optimal solution). **b** Impact of the MIS size on the observed deviation. The quality of the solution computed by the oscillators is a strong function of the MIS size. **c** Evolution of cluster diameter of the MIS vertices as a function of $V$ and $\eta$. **d** Effect of MIS size on the cluster diameter. The cluster diameter increases with the size of the MIS. Box plot results in **b**, **d**: center: median; box: interquartile range (IQR); whiskers: 1.5× IQR.

cardinality of the MIS which also increases with the size and the sparsity of the graph (Fig. 2b). Subsequently, Fig. 3b confirms the strong dependence of the quality of the solution on the size of the optimal MIS.

Next, we also explore the evolution of the cluster diameter (maximum phase difference between two oscillators in the same cluster) of the largest independent set in the observed phase ordering (Fig. 3c, d). The cluster diameter signifies the similarity

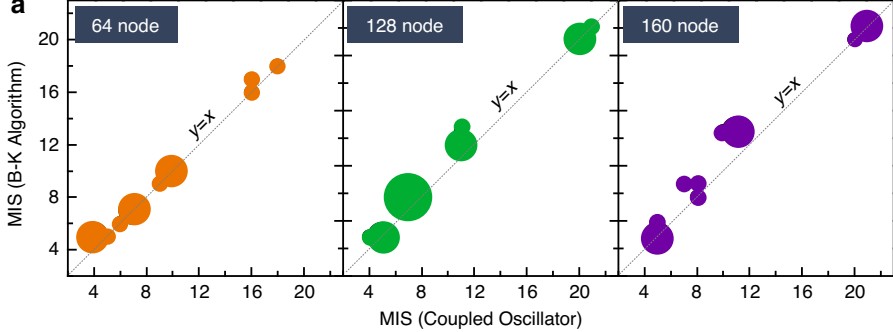

| b | DIMACS Graph | Node | Edge | MIS solution from oscillators (w/ post-processing) | MIS Solution from DIMACS |
|---|---|---|---|---|---|
| | 1tc_32 | 32 | 68 | 12 | 12 |
| | 1dc_64 | 64 | 543 | 10 | 10 |
| | 1tc_64 | 64 | 192 | 20 | 20 |
| | 1et_64 | 64 | 264 | 18 | 18 |
| | 2dc_128 | 128 | 5173 | 5 | 5 |
| | 1dc_128 | 128 | 1471 | 16 | 16 |
| | 1et_128 | 128 | 672 | 28 | 28 |
| | 1zc_128 | 128 | 2240 | 18 | 18 |
| | 2dc_256 | 256 | 17,183 | 6 | 7 |

**Fig. 4 Scalability of coupled oscillator approach. a** Bubble plots comparing the MIS solution obtained from the coupled oscillators (post expansion step) relative to the optimal solution (obtained from the B–K algorithm). **b** Graph instances from the DIMACS implementation challenge solved using coupled oscillators; the oscillators compute the optimal MIS solution in all except one graph.

of the eigenvalues of the nodes in the independent set cluster (see Supplement S1). Similar to the optimality trend observed in Fig. 3a, b, the cluster diameter increases with the sparsity of the graphs and shows a strong sensitivity to the size of the MIS (Fig. 3d). We note that the larger standard deviation observed in Fig. 3c, d can be attributed to the fact that the graphs considered in the analysis were randomly generated, and thus, have an arbitrary connectivity pattern. However, similar trends are also observed in $k$-nearest-neighbor connected graphs ($k = 4$ here) as shown in the Supplementary Note 3. Thus, both the computational characteristics of the hardware (i.e. cluster diameter) and the corresponding solution reveal that the oscillator system finds it more challenging to solve the larger and sparser graphs having a larger MIS—a validation of the hardness of the problem. In the context of the oscillator hardware, it is likely that the oscillators settle into one of the many local minima (corresponding to a sub-optimal solution) that is energetically close to the global minimum[65].

We also evaluate using simulations, the possibility of scaling our oscillator approach to compute the MIS in larger graphs (>30 nodes). Using a virtual coupled oscillator platform implemented in Xyce[66], we analyze: (a) randomly generated graph instances with 64, 128, 160 nodes having a wide range of connectivity ($\eta = 0.2, 0.4, 0.6, 0.8$) (Fig. 4a); three graphs are analyzed for each ($V$, $\eta$); and (b) some graph instances from the DIMACS[67] database up to 256 nodes (Fig. 4b). The simulations reveal that for larger graphs (specifically with high sparsity), the oscillators can yield a lower quality sub-optimal MIS solution; we observe empirically that the phase sequence tends to omit a few nodes of the optimal MIS solution. We, therefore, implement a simple scheme of expanding the largest observed independent set from the phase sequence to achieve a significant improvement in the MIS solution. The proposed post-processing scheme is discussed in

detail in the Supplementary Note 4. As revealed in Fig. 4a we observe that with post-processing, we achieve near-optimal/ optimal MIS solutions in the randomly generated graphs; optimal MIS solutions are achieved in 64% of the graphs, and solutions that are sub-optimal by up to one node are observed in ~90% of the analyzed graphs (similar to those observed in experiment). A comparison with other computational approaches for solving the MIS problem is shown in Supplementary Note 5. Furthermore, the oscillators compute an optimal solution in all except one of the DIMACS graphs analyzed here. This suggests that the coupled oscillator-based computing approach can be potentially scaled further through hardware-algorithm co-design although the effect of noise and the implementation and optimization of the coupling architecture will play a crucial role in system scalability.

## Discussion

The coupled oscillator platform demonstrated here facilitates a potentially scalable non-Boolean approach to problems that are considered computationally hard to solve on digital platforms. Our work demonstrates, using experiment and simulation, that the rich spatio-temporal dynamics of the coupled oscillators can be leveraged to compute (near-) optimal solutions to challenging optimization problems such as computing the MIS of a graph. Furthermore, since other hard optimization problems such as maximum clique[68], minimum vertex cover[69], minimum vertex coloring[70] can be reduced to the MIS problem, our work provides a pathway to the realization of a non-Boolean hardware accelerator for a broader class of computationally challenging optimization problems.

## Data availability

The datasets generated during and/or analyzed during the current study are available from the corresponding author on reasonable request.

## Code availability

All codes used in this work are either publicly available or available from the authors upon reasonable request.

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

## Author contributions

A.M. and S.J. designed the chip. A.M., D.S.T., and S.J. designed the experimental setup. M.K.B. performed the simulations. S.J., B.H.C., and N.S. supervised the study. A.M., N.S. wrote the manuscript. All authors discussed the results and commented on the manuscript.

## Competing interests

The authors declare no competing interests.
