## [Peer Review File · Nature Communications]

Reviewers' Comments:

Reviewer #1:

Remarks to the Author:

I am satisfied with the changes made by the authors in response to my comments and I recommend the paper for publication.

Reviewer #2:

Remarks to the Author:

The authors have done an excellent job in answering questions from the first round of reviews. Earlier works on using neural networks for computationally hard problems is properly credited, and limitations of the present approach are clearly explained. The work avoids presenting oscillators as a panacea for all challenges of modern computing hardware. The work presents an efficient analog circuit, in size that is well beyond a toy demonstration and also beyond the usual Boolean or neuromorphic constructions.

There are issues that still remain open: the effect of noise on scalability and also, I fear that the requirement for sparse connectivity will be restrictive. Despite these issues the paper could possibly be a major contribution for the field of analog circuit design and define new application areas for analog circuits.

Response Letter

We thank the editor and the reviewers for critically examining the manuscript and providing constructive comments towards its enhancement. A point-by-point response to the referees' comments is attached below.

Reviewer #1

I am satisfied with the changes made by the authors in response to my comments and I recommend the paper for publication.

Response:

We are thankful to the reviewer.

Reviewer #2

The authors have done an excellent job in answering questions from the first round of reviews. Earlier works on using neural networks for computationally hard problems is properly credited, and limitations of the present approach are clearly explained. The work avoids presenting oscillators as a panacea for all challenges of modern computing hardware. The work presents an efficient analog circuit, in size that is well beyond a toy demonstration and also beyond the usual Boolean or neuromorphic constructions.

There are issues that still remain open: the effect of noise on scalability and also, I fear that the requirement for sparse connectivity will be restrictive. Despite these issues the paper could possibly be a major contribution for the field of analog circuit design and define new application areas for analog circuits.

Response:

We thank the reviewer for the constructive feedback. We agree with the reviewer that the effect of noise and connectivity will play a crucial role in the system scalability and require optimization. We have therefore added the following discussion to the updated manuscript (in bold):

Corresponding updates to the main text (*Pg. 12 of the main text*)

“This suggests that the coupled oscillator-based computing approach can be potentially scaled further through hardware-algorithm co-design **although the effect of noise and the implementation and optimization of the coupling architecture will play a crucial role in system scalability**”